# Pt Modified Sb_2_Te_3_ Alloy Ensuring High−Performance Phase Change Memory

**DOI:** 10.3390/nano12121996

**Published:** 2022-06-10

**Authors:** Yang Qiao, Jin Zhao, Haodong Sun, Zhitang Song, Yuan Xue, Jiao Li, Sannian Song

**Affiliations:** 1The Microelectronic Research & Development Center, Shanghai University, Shanghai 200444, China; yangqiao@shu.edu.cn (Y.Q.); shd_1013484830@shu.edu.cn (H.S.); 2State Key Laboratory of Functional Materials for Informatics, Shanghai Institute of Microsystem and Information, Chinese Academy of Sciences, Shanghai 200050, China; zhaojin@mail.sim.ac.cn (J.Z.); ztsong@mail.sim.ac.cn (Z.S.); 3University of Chinese Academy of Sciences, Beijing 100049, China; 4Department of Mechatronics Engineering and Automation, Shanghai University, Shanghai 200444, China

**Keywords:** phase change memory, phase change material, high speed, thermal stability

## Abstract

Phase change memory (PCM), due to the advantages in capacity and endurance, has the opportunity to become the next generation of general−purpose memory. However, operation speed and data retention are still bottlenecks for PCM development. The most direct way to solve this problem is to find a material with high speed and good thermal stability. In this paper, platinum doping is proposed to improve performance. The 10-year data retention temperature of the doped material is up to 104 °C; the device achieves an operation speed of 6 ns and more than 3 × 10^5^ operation cycles. An excellent performance was derived from the reduced grain size (10 nm) and the smaller density change rate (4.76%), which are less than those of Ge_2_Sb_2_Te_5_ (GST) and Sb_2_Te_3_. Hence, platinum doping is an effective approach to improve the performance of PCM and provide both good thermal stability and high operation speed.

## 1. Introduction

In the past decades, rapid advances in artificial intelligence [1,2], supercomputing [3], and big data [4] have required ever−faster data exchange. While traditional hard disk drives and solid−state drives struggle to meet demand, new types of memory have taken the challenge. Phase change memory (PCM) is considered a promising non−volatile memory technology due to its advantages of high speed, high density, high scalability, low operating voltage, and high endurance [2,5,6,7,8]. As the storage medium of PCM, phase change material can achieve reversible phase transitions between crystalline and amorphous states under the action of electrical pulses. The memory relies on the resistance difference between the crystalline and amorphous states of phase change materials to store “0” and “1” [9,10,11,12]. The common phase change material Ge_2_Sb_2_Te_5_ (GST) is currently the most successful commercialized material. However, poor 10−year data retention (~85 °C), slow operating speed (~20 ns), and a density change rate of 6.5% limit its wider application in electrical devices [13,14]. Therefore, looking for a phase change material with high amorphous thermal stability and fast speed is the key to improving the performance of PCM [2,15,16].

The PCM device based on Sb_2_Te_3_ shows fast operation speed. However, the low crystallization temperature (<100 °C) makes the amorphous state unstable, which means that Sb_2_Te_3_ is not suitable for PCM application. Doping is a good way to improve thermal stability and speed. Some researchers have obtained high−performance phase change materials by doping Sb_2_Te_3_, such as Sc_0.2_Sb_2_Te_3_. It achieved an ultra−fast operation speed of 700 ps and the data retention of ~87 °C [5], which satisfies the requirements of subnanosecond high−speed cache memory. However, in some applications filed [13,14], higher data retention is required. Since the thermal stability of materials is related to data retention, we need to find a phase change material with high thermal stability. The traditional precious metals materials (Au, Ag, Pt) have excellent chemical stability and are conducive to engineering applications. Our selection principle is that the element with high electronegativity is used as the doped element, so as to form a stable chemical bond with the elements of the parent material to ensure no phase separation during the operation of the device. Silver was, therefore, rejected as a candidate material. At the same time, considering the cost of gold and platinum, platinum is finally selected as the dopant.

In this work, we have performed electrical tests based on Pt−Sb_2_Te_3_ devices and microscopic characterization of films. The PCM devices based on Pt_0.14_Sb_2_Te_3_ (PST) show fast operation speed, high data retention, and good endurance. Meanwhile, the corresponding microstructure of PST explains the origin of its high performance.

## 2. Materials and Methods

### 2.1. Film Preparation and Testing

The Sb_2_Te_3_, Pt_0.1_Sb_2_Te_3_, Pt_0.14_Sb_2_Te_3_ (PST), and Pt_0.22_Sb_2_Te_3_ films are deposited by sputtering of Pt and Sb_2_Te_3_ targets. The compositions of these films were measured by energy−dispersive spectroscopy (EDS). Films with a thickness of 200 nm were deposited on SiO_2_/Si (100) substrates for resistance−temperature (R−T) and X−ray diffraction (XRD) tests. In situ R−T measurement was conducted by a homemade vacuum heating table, and the heating rate was 20 °C /min. The film was heated in a vacuum chamber with a heating rate of 60 °C/min, and the isothermal change in resistance with increasing temperature was recorded to estimate the 10−year data retention. The X−ray reflectivity (XRR) experiment (Bruker D8 Discover) was used to test the density change of films before and after crystallization. X−ray photoelectron spectroscopy (XPS) experiment was used to evaluate the bonding situation. Then the film (about 20 nm) was deposited on the ultra−thin carbon film, and its microstructure was studied by Transmission Electron Microscope (TEM). TEM is manufactured by Hitachi Limited in Tokyo, Japan.

### 2.2. Device Fabrication

T−shaped PCM devices were prepared by 0.13 μm complementary metal−oxide semiconductor technology. The diameter of the tungsten bottom electrode is about 60 nm. The 70 nm−thick phase change material and 20 nm−thick TiN as adhesion layer were deposited through the sputtering method over a 60 nm diameter of tungsten heating electrode. The device is measured by the Keithley 2400 C source meter and Tektronix AWG5002B pulse generator. The Keithley 2400 C source meter and Tektronix AWG5002B pulse generator are manufactured in the Beaverton, OR, United States by Tektronix.

## 3. Results

### 3.1. Improved Device Performance

The films with different Pt compositions were performed by resistance−temperature (R−T) tests, as shown in Figure 1b. The R−T curves show that doping Pt into Sb_2_Te_3_ can enhance the crystallization temperature of the material, and the crystallization temperature increases with more Pt. The amorphous resistance of the material first increases and then decreases with the content of Pt. This is due to the low crystallization temperature of as−deposited Sb_2_Te_3_ film and partly crystallization, which will be confirmed by subsequent XRD experiments. Dopant atoms can increase scattering probability, so the effect of scattering is enhanced as the doping concentration increases and results in an increase in resistivity. However, when the doping concentration is too high, the metallicity of the material increases and the resistivity decreases. The crystallization temperature can be measured via Raman or XRD measurements and is simply approximated by the curve of resistivity. In this paper, we chose to use the R−T curve to calculate the crystallization temperature. In the R−T diagram, the crystallization temperatures of Pt_0.1_Sb_2_Te_3_, Pt_0.14_Sb_2_Te_3_ (PST), and Pt_0.22_Sb_2_Te_3_ are 137 °C, 199 °C, and 236 °C, respectively, which indicates that the thermal stability of the Sb_2_Te_3_ alloy is improved after Pt doping. The resistance of the PST drops by more than an order of magnitude, which is enough to distinguish the ON/OFF states used in the PCM storage devices. Therefore, we believe that the performance of the PST film is greatly improved. Figure 1c shows the resistance time (R−T) curve. The 10−year data retention can be estimated by the Arrhenius equation:(1)$$t=\tau exp\left(E_{a}/K_{B}T\right)$$

The 10−year data retention for GST and PST are expected to be 85 °C and 104 °C, respectively, with corresponding activation energies (*E_a_)* of 2.57 eV and 1.86 eV. The activation energy errors are 0.05 eV and 0.40 eV, respectively. We find that 10−year data retention of PST films is higher than that of most phase change memories, such as GST (~85 °C) and SST (~87 °C) [5].

Accordingly, based on standard 0.13 μm complementary metal−oxide semiconductor (CMOS) technology, T−shape PCM devices based on PST were fabricated, as shown in Figure 1a. Then, the electrical properties of the device are characterized. RESET, SET, and READ functions can be realized by using different pulse waveforms. Figure 1d shows the SET−RESET windows using the resistance−voltage (R−V) curves. The high/low resistance ratio (R_RESET_/R_SET_) is about two orders of magnitude, which can meet the requirement of the ON/OFF ratio used in PCM. When the voltage pulse width of 6 ns, the SET/RESET voltage of the PST device requires 1.2 V/3.8 V. However, GST requires 4.6 V/5.5 V with a 10 ns operation speed [5]. A pre−program voltage applied by pre−operation to GST enables a SET speed of 500 ps in a restricted device structure [17]. This competitive recording speed is already comparable to DRAM and SRAM (1−10 ns) [18]. As shown in Figure 1e, the endurance period is revealed after we alternately apply two appropriate SET and RESET voltage pulses. Figure 1e shows that the reversible phase transition characteristic is up to 5 × 10^5^ switching cycles with a resistance ratio of two orders of magnitude. The switching cycles and resistance ratio of PST are better than Sb_2_Te_3_ [19]. The endurance performance is higher than GST [20] using the T−shaped device structure. All above, compared with GST, faster operation speed and better endurance of PST have proved Pt doping Sb_2_Te_3_ with suitable composition is a promising novel phase−change material. 

### 3.2. Characterization of Thin Film Structure

The XRD method was employed to characterize the lattice structure of PST film. Figure 2a,b shows the XRD results of PST and Sb_2_Te_3_ films at different annealing temperatures. The diffraction peak of Sb_2_Te_3_ appears in the deposited state, indicating that the deposited Sb_2_Te_3_ has crystallized. At this time, there is no diffraction peak of PST, so the PST has not crystallized. At 200 °C, the FCC phase appeared in the PST, which indicated that Pt inhibited the formation of the FCC phase and increased the crystallization temperature. When the annealing temperature is 260 °C, both PST and Sb_2_Te_3_ have only the diffraction peaks of the hexagonal phase. Compared with pure Sb_2_Te_3_ film, the diffraction peaks of PST film become wider, the intensity of the peak becomes lower, and some diffraction peaks disappear. In addition, a difference in the full width at half maximum (FWHM) of the diffraction peak is observed on the XRD curves. According to the Scherrer formula:(2)β=Kλ/L(cosθ)

*K* in the equation is the Scherrer constant (*K* = 0.89), *β* is the grain size, *L* is the full width at half maximum (FWHM) of the diffraction peak of the sample, *θ* is the diffraction angle, and *λ* is the X−ray wavelength (0.154056 nm). The FWHM of PST was significantly higher than that of Sb_2_Te_3_, indicating that the incorporation of Pt inhibited the crystallization growth process, and grain refinement was obvious. Reducing grain size is ideal for programming areas [21].

To study the crystalline phase and grain size more intuitively, high−resolution transmission electron microscopy (HRTEM) and the associated selected area electron diffraction (SAED) patterns for Sb_2_Te_3_ film and PST films are presented in Figure 3. In total, 2 samples were annealed at the temperature of 260 °C for 5 min. The annealed films are both in a polycrystalline state. Comparing Figure 3a,b, it can clearly be seen that the grain size decreases significantly. In Figure 3c, doping Pt reduces the grain size of Sb_2_Te_3_ from 50 nm to about 5~10 nm, which confirms that the half−height width of PST is much larger than that of Sb_2_Te_3_. Meanwhile, according to Figure 3a,b, small crystal grains of the PST film can be also inferred from the continuous diffraction rings [22]. Smaller grain size increases the surface volume ratio, thus generating more grain boundaries [23]. As the number of grain boundaries increases, the crystal diffusion and slippage can be reduced. Hence, the residual stress in the bulk of films can be degraded [24,25]. Moreover, the increased grain boundaries provide a phonon and electron scattering center, and the decreased thermal and electrical conductivity will improve the energy efficiency of the Joule heating [26]. According to the HRTEM image in Figure 3d,e, the crystal structure is in the hexagonal phase after the calculation of inter−planar distance. They all belong to the (1010) and (105) families, which indicates the crystalline state of the PST film is composed of the hexagonal phase. The result of SAED in Figure 3b matches the HRTEM perfectly. In other words, Pt doping affects the crystallization behavior of the Sb_2_Te_3_ film without forming any new phase or structure.

Crystallization usually leads to an increase in film density and a reduction in film thickness. The information on the density change upon crystallization is of paramount importance in phase change media technology since it is related to the stresses induced in the system during the write/erase cycle. The change of density before and after the phase transition of the sample was measured by XRR. Figure 4 separately depicts the XRR curves of PST films in amorphous and crystalline states. Based on the peak position shift, a linear fit calculation is performed, as shown in Figure 4b. During the transition from the amorphous to the crystalline state, the thickness change rate of PST film is only 4.7%, while the thickness change rate of Sb_2_Te_3_ and GST films are 7.5% [27] and 6.5%, respectively. This enhancement is responsible for the improved cyclability.

### 3.3. Evidence of Pt Occupying Positions

Experiments have proved that when element B is replaced by element C and bonded with element A, if the electronegativity of element C is greater than that of element B, the binding energy of element A increases [28]. In Figure 5a,b, the binding state of Sb_2_Te_3_ and PST is revealed by XPS. When the Pt atom enters Sb_2_Te_3_, if the Pt atom replaces the Sb atom and combines with the Te atom, since the electronegativity of Pt (2.2) is higher than that of Sb (2.05) and Te (2.12), the binding energy of Te will shift towards the high binding energy, which is consistent with the phenomenon in the experiment in Figure 5. Combined with the XRD result that shows there is no new phase, this confirms that Pt replaces the position of Sb.

## 4. Conclusions

In this work, we systematically studied the performance of PST. The PCM devices based on PST can achieve higher speed and data retention than GST devices. According to XPS and TEM analyses, the microstructure feature of Pt−modification Sb_2_Te_3_ film is explained clearly. The reduced grain size and formation of Pt−Te bonds are the main reasons for the improved properties. Subsequently, a boost in device endurance gave the credit to the reduced density change rate. The improvement of these properties is conducive to the commercial application of the material. Such experimental results show that PST has broad application prospects in complex environments.

## Figures and Tables

**Figure 1 nanomaterials-12-01996-f001:**
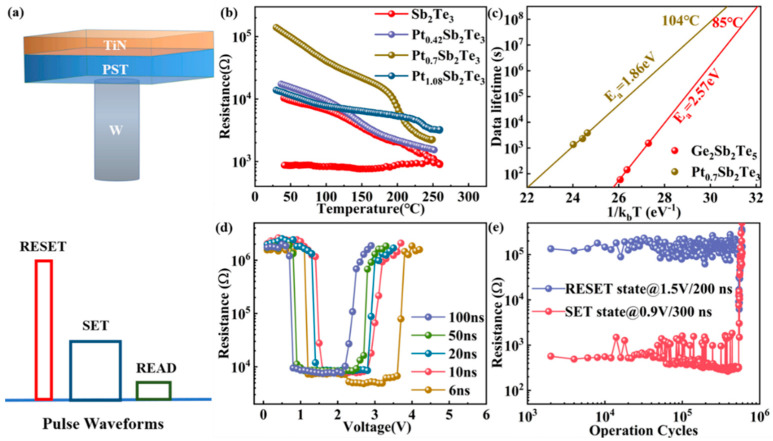
Device performance. (**a**) The schematic diagram of the T−shaped phase change memory (PCM) device. Schematic diagram of three pulse voltages of RESET, SET, and READ of PCM. (**b**) The temperature dependence of the resistance of Sb_2_Te_3_, Pt_0.1_Sb_2_Te_3_, Pt_0.14_Sb_2_Te_3_ (PST), and Pt_0.22_Sb_2_Te_3_ films at the same heating rate of 20 °C/min. (**c**) At the heating rate of 60 °C/min, the extrapolated fitting line based on the Arrhenius formula shows the 10−year data retention temperature and crystallization activation energy. (**d**) Resistance−voltage characteristics of PST based T−shaped PCM device. The SET−RESET programming windows are obtained under different pulse widths. (**e**) Endurance characteristic of PST based PCM T−shaped devices.

**Figure 2 nanomaterials-12-01996-f002:**
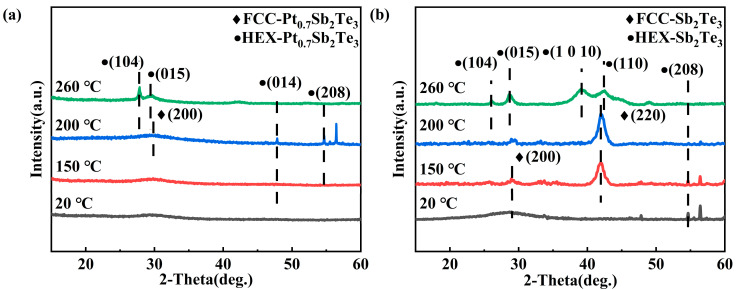
XRD results of the Sb_2_Te_3_ and PST. (**a**,**b**) XRD curves of PST and Sb_2_Te_3_ films were annealed at 150 °C, 200 °C, and 260 °C for 5 min in an N_2_ atmosphere.

**Figure 3 nanomaterials-12-01996-f003:**
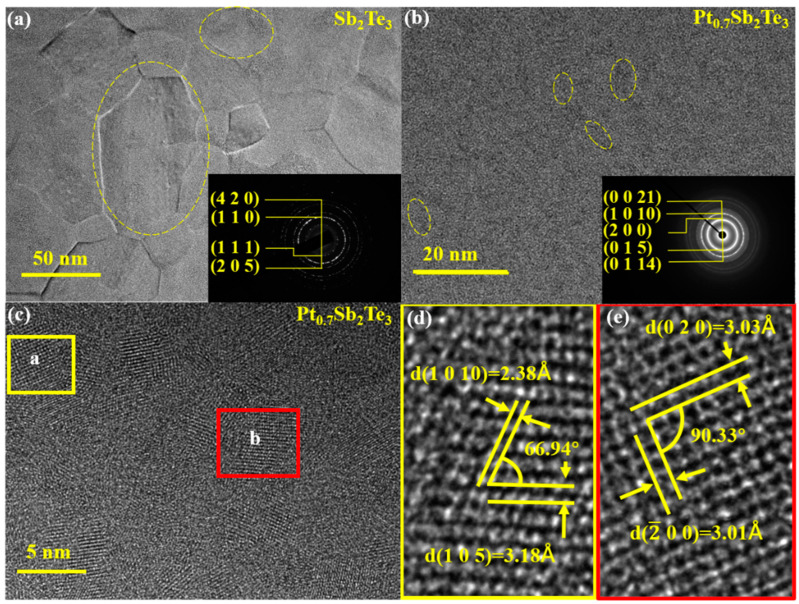
(**a**) TEM image of Sb_2_Te_3_ film after annealed at 260 °C. (**b**) TEM image of PST film after annealed at 260 °C. (**c**–**e**) HRTEM images of PST film after annealed at 260 °C.

**Figure 4 nanomaterials-12-01996-f004:**
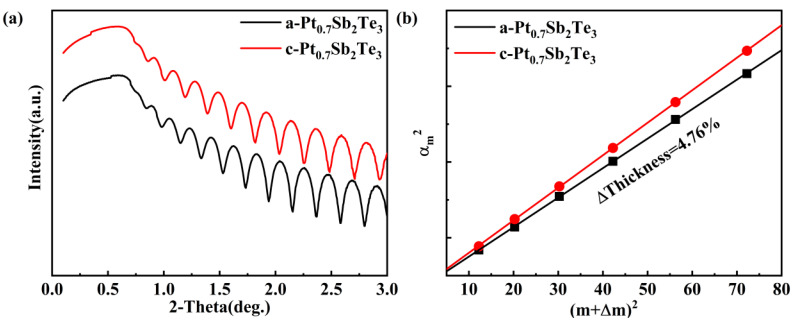
The density−change rate before and after PST crystallization (**a**) XRR curves of amorphous and crystalline PST films. (**b**) Bragg fitting curves of amorphous and crystalline films.

**Figure 5 nanomaterials-12-01996-f005:**
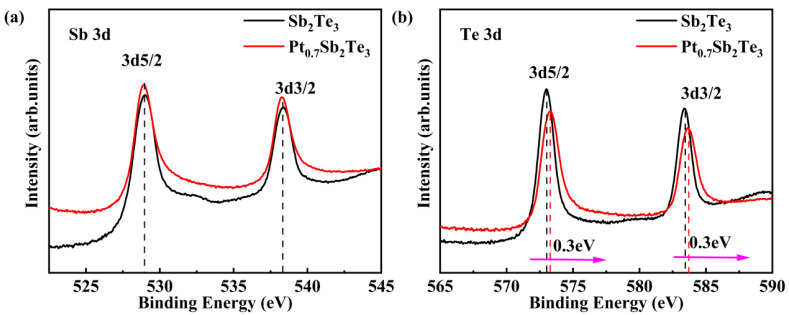
XPS spectra of Sb_2_Te_3_ and PST films annealed at 260 °C (**a**) Sb 3d and (**b**) Te 3d.

## Data Availability

Not applicable.

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
