# Peer review of "Pt Modified Sb2Te3 Alloy Ensuring High−Performance Phase Change Memory"

_nanomaterials, 2022, doi:10.3390/nano12121996_

Round 1
Reviewer 1 Report
The paper discusses a doping scheme for enhancing the endurance of Sb2Te3 PCMs for high speed phase change memories, complete with metrology and physical explanations. While schemes of doping are not new, this represents a reasonable jump in technology that warrants consideration.
Some comments:
- Line 53: “At the same time, considering the 53 cost, platinum is finally selected as the dopant”. Since Pt is considerably more expensive than silver the authors must clarify that this statement relates to gold, not silver.
- Line 93: The crystallization temperature can be really measured via Raman or XRD measurements and is simply approximated by the curve of resistivity. The statement should be amended to reflect that.
- General point: Stability is often defined by the idea of “data retention” which might be related to the temperature, but they are not the same. This is stated on line 101, but perhaps it would e better to state on lines 46,96 or at least mention that fact prior to line 101.
- The equation on line 101 seems to be an image instead of an actual typeset equation
- Loke et al have shown 500 ps speed GST based PCM/PCRAM via pre-biasing, which should be mentioned (Breaking the Speed Limit of Phase Change Memory, Science, 2012) in line 114 that mentions only doping schemes for breaking the speed limits of PCRAMs
Apart from those minor comments, the paper is clean, the language is fairly good, and results convincing. I recommend publishing if the authors address the minor comments above.
Reviewer 2 Report
This work presents a useful study of phase change devices based on Pt doped Sb2Te3. This is an interesting subject and the results appear to be generally sound. I would recommend publication after the authors address the following:
(1) Fig. 1c shows a huge extrapolation that makes it hard to see the actual quality of the fits. This is especially for GST, but also true for the new material. I would suggest changing the scales and plotting in way that allows one to understand how good the fit is. Some statement about the uncertainty in the activation energy is needed.
(2) In the pulse waveforms of Fig 1(a) the time scale should be shown. Also it should be commented on. The higher crystallization temperature should affect the needed pulses as compared to GST which should be discussed. In the regard switching times are useful to know.
(3) Some statements don't have clear meaning. "The electrons in the nonlocal state are transformed into local state, resulting in more electron scattering." (this seems meaningless, what does localization have to do with causing scattering?). Also it seems strange to say "The metallicity is enhanced, which leading to the reduced amorphous resistance of the material." (seems like they are saying the resistance is reduced leading to reduced resistance, which is meaningless)
(4) The XRD instrument is probably "Bruker" not "Burker".
